# Evolution Characterization and Pathogenicity of an NADC34-like PRRSV Isolated from Inner Mongolia, China

**DOI:** 10.3390/v16050683

**Published:** 2024-04-26

**Authors:** Hong-Zhe Zhao, Chun-Yu Liu, Hai Meng, Cheng-Long Sun, Hong-Wen Yang, Hao Wang, Jian Zou, Peng Li, Feng-Ye Han, Gen Qi, Yang Zhang, Bing-Bing Lin, Chuang Liu, Meng-Meng Chen, Pan-Ling Zhang, Xiao-Dong Chen, Yi-Di Zhang, Qian-Jin Song, Yong-Jun Wen, Feng-Xue Wang

**Affiliations:** 1Key Laboratory for Clinical Diagnosis and Treatment of Animal Diseases of Ministry of Agriculture, College of Veterinary Medicine, Inner Mongolia Agricultural University, Hohhot 010018, China; hongzhezhao@163.com (H.-Z.Z.); menghai19560856897@163.com (H.M.); scl17615003714@163.com (C.-L.S.); 15228701650@163.com (H.-W.Y.); 17684721857@163.com (H.W.); jianzou0817@163.com (J.Z.); eqli159@126.com (P.L.); han1292656694@163.com (F.-Y.H.); 15848114044@163.com (G.Q.); 15147255058@163.com (Y.Z.); 15248097007@163.com (B.-B.L.); 18724258051@163.com (C.L.); cmm1284742747@163.com (M.-M.C.); zhangpl_1223@163.com (P.-L.Z.); 13190576177@163.com (X.-D.C.); 17610105789@163.com (Y.-D.Z.); yongjunwen@126.com (Y.-J.W.); 2Medical Experiment Center, Inner Mongolia Medical University, Hohhot 010018, China; chunyuliu_vet@163.com; 3Yinchuan Animal Husbandry Technology Extension Service Center, Yinchuan 750000, China; 15848152770@163.com

**Keywords:** porcine reproductive and respiratory syndrome virus, genomic characterization, NADC34-like, pathogenicity

## Abstract

Porcine reproductive and respiratory syndrome virus (PRRSV) is a pathogen that causes severe abortions in sows and high piglet mortality, resulting in huge economic losses to the pig industry worldwide. The emerging and novel PRRSV isolates are clinically and biologically important, as there are likely recombination and pathogenic differences among PRRSV genomes. Furthermore, the NADC34-like strain has become a major epidemic strain in some parts of China, but the characterization and pathogenicity of the latest strain in Inner Mongolia have not been reported in detail. In this study, an NADC34-like strain (CHNMGKL1-2304) from Tongliao City, Inner Mongolia was successfully isolated and characterized, and confirmed the pathogenicity in pigs. The phylogenetic tree showed that this strain belonged to sublineage 1.5 and had high homology with the strain JS2021NADC34. There is no recombination between CHNMGKL1-2304 and any other domestic strains. Animal experiments show that the CHNMGKL1-2304 strain is moderately virulent to piglets, which show persistent fever, weight loss and high morbidity but no mortality. The presence of PRRSV nucleic acids was detected in both blood, tissues, nasal and fecal swabs. In addition, obvious pathological changes and positive signals were observed in lung, lymph node, liver and spleen tissues when subjected to hematoxylin–eosin (HE) staining and immunohistochemistry (IHC). This report can provide a basis for epidemiological investigations and subsequent studies of PRRSV.

## 1. Introduction

Porcine reproductive and respiratory syndrome (PRRS) is an infectious disease caused by the PRRSV, which was first reported in North America in 1987 and subsequently detected in Europe and Asia in 1990 [1,2,3,4]. The virus is currently endemic in most pig farming countries and is one of the main causes of economic losses in the industry [5]. The disease is reported to cause economic losses of about USD 560 million per year to the United States’ farming industry, with weaned piglets and breeding stock accounting for 55% of the total cost [6]. In comparison, there was an average economic loss of EUR 126 per sow during the PRRS outbreak in Europe [7]. PRRSV is an enveloped, single-stranded, positive-stranded RNA virus that belongs to the genus *Arterivirus* of the *Arteriviridae*, along with the equine arteritis virus (EAV), simian hemorrhagic fever virus (SHFV) and lactate dehydrogenase elevation virus (LDEV) [8]. In addition, the full-length PRRSV genome is approximately 15 kb and encodes a 5′ untranslated region (UTR), at least 11 open reading frames (ORF), a 3′ UTR and a 3′ poly(A) tail [9,10]. After the introduction of PRRSV-2 into our country, the strain has been mutating and recombining over time. The NADC34-like strain has now become the predominantly endemic strain in some regions [11,12]. Pathogenicity studies have been conducted on NADC34-like strains isolated from Heilongjiang, Tianjin and Jiangsu [13,14,15]. However, detailed studies on this newest strain in Inner Mongolia have never been reported. Therefore, this study isolated and identified a strain of NADC34-like virus with typical PRRSV characteristics from Tongliao City, Inner Mongolia, and pathogenicity experiments were carried out to provide a basis for PRRSV epidemiological investigations and subsequent research.

## 2. Materials and Methods

### 2.1. Sampling and Virus Isolation

In this study, lungs and blood were collected from sick piglets. The lungs were ground in phosphate buffered salt solution (PBS) using a mortar. The supernatant was then aspirated and filtered using a 0.22 μm filter. Total RNA was extracted using an RNA extraction kit (Fastagen Biotech, Shanghai, China), and cDNA was synthesized using a GoScript reverse transcription kit (Promega, Beijing, China). The samples that tested positive were inoculated with porcine alveolar macrophages (PAM) to isolate the virus. The PAM cells were obtained from lung lavage of 4-week-old piglets without a specific disease source and preserved in RPMI-1640 complete medium containing 10% fetal bovine serum. The virus was harvested 3 days after inoculation and stored in a −80 °C refrigerator.

### 2.2. Electron Microscopic Observation

Virulent strains were expanded in culture and then centrifuged at 3000× *g* for 20 min using a Millipore virus concentration column. The concentrated virus was then adsorbed on a copper grid for 5 min. The excess virus solution was absorbed on filter paper and stained with 2% phosphotungstic acid (pH = 7.0) for 30 s. The filter paper once again absorbed the excess dye solution. The morphological characteristics of the virus were observed using a transmission electron microscope (Hitachi, Tokyo, Japan).

### 2.3. Immunofluorescence Identification

PAM cells were fixed for 1 h at room temperature with paraformaldehyde at 48 h postinfection by PRRSV. Then, they were blocked with 5% BSA at 37 °C for 1 h. After that, the cells were washed three times with PBST for 5 min each. Next, the cells were incubated with antibodies against PRRSV M protein (GeneTex, Shenzhen, China) at 37 °C for 1 h, followed by three washes with Phosphate-Buffered Saline with Tween 20 (PBST) for 10 min each. Subsequently, the cells were incubated with goat anti-rabbit IgG H&L (Alexa Fluor 488) (Abcam, Cambridge, England) at 37 °C for 45 min, followed by three washes in PBST for 10 min each. After that, the cells were incubated with 4′,6-diamidino-2-phenylindole (DAPI) at room temperature for 20 min, followed by three washes with PBST for 5 min each. Finally, the results were observed using fluorescence inverted microscopy.

### 2.4. Primers’ Design and Amplification of Genomic Full Length

The full-length primers were designed using Primer5 and SnapGene based on the whole genome sequence of JS2021NADC34 (GenBank: MZ820388.1). The full-length PRRSV is approximately 15 kb in length, and 8 pairs of mutually overlapping specific primers will be designed to amplify the full-length sequence. The specific primer information and annealing temperatures are shown in Table 1. An RNA template (5 μL) was used to obtain cDNA in a total volume of 20 μL. The PCR reaction system consisted of 0.5 µL of LA Taq, 5 µL of buffer, 8 µL of dNTP, 1 µL each of upstream and downstream primers, 33.5 µL of water and 1 µL of template. The 35 cyclic conditions were as follows: denaturation at was 94℃ for 1 min, 98 °C for 10 s, annealing for 30 s, 72 °C for 3 min and 72 °C for 10 min. Once the PCR amplification was completed, the samples were individually loaded into 1% agarose gel wells and electrophoresed at 110 V for 35 min. The results were then observed using a gel imager and preserved. Finally, the PCR products were sent directly to Shanghai Biotech Co. Ltd. for Sanger sequencing.

### 2.5. Sequencing Analysis

Combine the sequencing results with the sequencing peak maps for analysis, and perform whole-genome assembly using SeqMan (version 7.1.0) software (DNASTAR Inc., Madison, WI, USA). Next, the homology between the gene of strain CHNMGKL1-2304 and genes from other strains was analyzed using Geneious (version 4.5.3) (Biomatters, Auckland, New Zealand) and MegAlign (version 7.1.0) software (DNASTAR Inc., Madison, WI, USA). Furthermore, a phylogenetic tree was constructed using the Mega (version 7.0.26) software (Mega Limited, Auckland, New Zealand). Finally, the genomic recombination analysis was performed with SimPlot (version 3.5.1) software.

### 2.6. Animals and Experimental Design

Six 4-week-old piglets without a specific source of infection were randomly divided into two groups of three piglets each. A total of 3 × 10^6^ TCID_50_ virus particles were inoculated intramuscularly and nasally sprayed per piglet in the challenge group. Three piglets in the control group were inoculated with 1 mL of DMEM in the same manner. Daily observations were made on clinical symptoms, temperature and body weight and were scored according to the clinical assignment values in Appendix A [16,17]. Whole blood, serum, nasal swabs and fecal swabs were collected at 0, 3, 5, 7, 10 and 14 days after the challenge. After 14 days, all piglets were euthanized for further pathological observations.

### 2.7. Determination of Viral Load in Serum, Tissue, Nasal and Fecal Swabs

RNA was extracted from serum, tissue, nasal and fecal swabs using an RNA extraction kit (Promega, Beijing, China). This extracted RNA was then used as a template for determining the viral load in serum, tissue, nasal and fecal swabs through fluorescent quantitative RT-PCR. The fluorescent quantitative RT-PCR reaction was performed with One-Step Probe RT-qPCR Kit (TransGen Biotech, Beijing, China). The RT-PCR method used in this study referred to the method developed by K. Wernike et al., with the addition of standards [18].

### 2.8. Antibody Detection Assay

Antibody titers were detected at 0, 3, 5, 7, 10 and 14 days post-challenge using the PRRSV N and GP5 protein ELISA antibody detection kit (Keqian and Yaji, Wuhan and Shanghai, China). According to the kit instructions, when the KO value is greater than or equal to 40, it is judged to be positive for anti-N protein antibody. Meanwhile, when the OD450 value of the sample was greater than 0.192, it was determined to be positive for anti-GP5 protein antibody.

### 2.9. Histopathology and Immunohistochemical Staining

Tissue samples of the heart, liver, spleen, lungs, kidneys, lymph nodes, brain and stomach were taken during autopsy. These samples were fixed in 10% formalin and subjected to HE staining and IHC observation. The antibody used for immunohistochemistry was the PRRSV M protein antibody (GeneTex, Shenzhen, China). We applied Image-Pro Plus 6.0 image analysis software to calculate the number of weak-medium-strong positive cells in the field of view and then performed the H-score [19]. The H-score is a value between 0 and 300, and a larger value indicates a stronger integrated positive signal.

## 3. Results

### 3.1. Results of Virus Isolation and Identification

Positive samples were inoculated with PAM and Marc-145 cells, respectively. The figure below clearly shows that PAM cells are visibly aggregated and fragmented after 72 h (Figure 1A), while the control cells appear normal (Figure 1D). Similarly, Marc-145 cells exhibit aggregation and ridge-like structures at 96 h after infection with the virus (Figure 1B), while control cells remain normal (Figure 1E). We conducted IFA using an antibody to the PRRSV M protein against CHNMGKL1-2304, which revealed a significant area of green fluorescence in the cytoplasm (Figure 1C), while no fluorescence was observed in the control (Figure 1F). Finally, we concentrated the virus and performed phosphotungstic acid staining, which allowed us to clearly observe the presence and size of the virus particles under electron microscopy (Figure 1G,H). The size of PRRSV was found to be between 50 and 70 nm, indicating successful isolation of the strain.

### 3.2. Amplification and Sequencing of Full Genome 

PCR was performed according to the designed primers. The whole genome of the CHNMGKL1-2304 strain was successfully amplified with eight overlapping fragments (Appendix A: A–H). The negative control (Appendix A: I without primers) was not amplified. The sizes of the A–H target genes were 335 bp, 2591 bp, 2646 bp, 2350 bp, 2758 bp, 2758 bp, 2573 bp and 1047 bp, respectively. The sizes of the target genes were consistent with the expected sizes, indicating more accurate results. By Sanger sequencing of splicing and assembly, we obtained a full-length genome of size 15,112 nt (Appendix A). The full gene sequence was subsequently submitted to the GenBank database (GenBank number: OR753369.1).

### 3.3. Comparison Results of Homologous Sequences

In this study, we obtained the genes of each part of the whole genome of strain CHNMGKL1-2304 using Geneious (version 4.5.3) software and compared the homology using DNAStar (version 7.1.0) software. We selected strains of PRRSV lineage 1, 3, 5 and 8 as well as CHNMGKL1-2304 for comparison. In the previous identification, we found a high homology between strains CHNMGKL1-2304 and JS2021NADC34. Therefore, we also compared them for each gene, and the results are shown in Table 2. We discovered that this strain has high homology with all parts of the genome of JS2021NADC34, with homology levels above 94%. It also has high homology with IA/2014/NADC34 found in the United States. After comparison, we determined that the strain belongs to sublineage 1.5 and did not recombine with other lineages. To investigate whether this strain undergoes inter-spectral recombination with locally occurring NADC34-like strains, we selected 10 locally reported NADC34-like strains for homology comparison in this study. We found that the CHNMGKL1-2304 strain showed high homology (88.4–94.4%) with locally reported NADC34 (Table 3). We also found that NSP1 and NSP2 were not conservative and prone to mutation. Based on these results, we initially determined that the strain did not undergo recombination. However, further software analysis is necessary.

### 3.4. Results of Reorganization Analysis

In this study, we utilized the findings from our previous analysis to select whole genome sequences of strains that exhibited high homology in each section. We then analyzed these sequences using RDP (version 4) software and employed seven recombination detection algorithms (RDP, GENECONV, BootScan, MaxChi, Chimaera, SiScan and 3Seq) to identify strains that could potentially be recombinant with the CHNMGKL1-2304 strain. Additionally, we employed SimPlot (version 3.5.1) software to identify the recombination breakpoints. The strain JS2021NADC34 demonstrated significant similarity to the strain CHNMGKL1-2304. Although the CHNMGKL1-2304 strain has the highest similarity to the strain JS2020 at 2 loci, it also has high homology to the strain JS2021NADC34 at this locus. Consequently, we concluded that there was no recombination between the CHNMGKL1-2304 strain and any other strains (Appendix A).

### 3.5. Phylogenetic Analysis Based on ORF5 Gene and Full-Length Genome

To understand the evolutionary characteristics of the isolated strains in Inner Mongolia, a phylogenetic tree was constructed based on the ORF5 gene and the whole genome of PRRSV in this study (Figure 2). From Figure 2A, it is evident that the NADC34-like strains isolated in Inner Mongolia belong to the same branch and are closely related. The whole genome phylogenetic tree, shown in Figure 2B, reveals that the CHNMGKL1-2304 strain isolated in Inner Mongolia has a closer homology and forms an independent branch with the NADC34-like strains isolated in different provinces in China. Moreover, this strain shares a close homology with the highly pathogenic strain JS2021NADC34, which was isolated from Jiangsu in 2021. This suggests that the highly pathogenic strain found in Inner Mongolia likely originated from Jiangsu. Subsequently, we divided the NADC34-like strains isolated from 2017 to 2023 into three groups. We observed that the branches of group 3 gradually became larger over time and showed a tendency to form unique branches. On the other hand, strains from group 1 and group 2 are clearly becoming less prevalent (Figure 2B). This finding provides new insights into the development trend of the epidemic.

### 3.6. Clinical Symptom of Piglets Infected with CHNMGKL1-2304

Clinical symptoms of CHNMGKL1-2304-infected piglets mainly include persistent high fever, anorexia, recumbency, slight coughing, shortness of breath, redness, trembling, paralysis and weight loss. This strain causes a rapid onset of disease in pigs, with the clinical score increasing 1 day after infection (Figure 3A). This is evidenced by a rapid rise in body temperature to about 41 °C (Figure 3B), anorexia and, in one case, diarrhea, but with normal consciousness. After the challenge, the overall clinical score gradually increased and stabilized (Figure 3A). The temperature of piglets in the challenge group remained consistently high from 6 to 14 days post-challenge (dpc) (Figure 3B), while the control group had a normal temperature. Additionally, piglets in the challenge group gained an average of 0.196 kg/day throughout the experiment compared to 0.4 kg/day in the control group (Figure 3C). Three days after the challenge, one of the piglets in the experimental group became paralyzed. No piglets died during the entire trial period.

### 3.7. Viral Load and Antibody Level

Using the PRRSV Taqman absolute fluorescence quantitative PCR method, serum, tissue, nasal and fecal swabs were collected from the experimental and control groups to determine the PRRSV genome copy number. It was discovered that piglets in the experimental group had detectable levels of the CHNMGKL1-2304 strain in their blood 3 days after being challenged with the virus. The viral load increased gradually after 3 days, peaking at 10 dpc, and then declined. No PRRSV genome was found in the control group (Figure 4A). Upon dissecting the tissues at 14 dpc, higher viral loads were observed in the lungs and hilar lymph nodes of the experimental group, which correlated with visible pathological changes (Figure 4B). The lowest viral loads were found in the stomach of the experimental group, and no viral genome was detected in any of the tissues from the control group. In the nasal and fecal swabs of both the experimental and control groups, the viral genome was detected at 3 dpc and gradually decreased over time. This suggests that the CHNMGKL1-2304 strain can be shed in infected piglets as early as 3 days, allowing for the spread of the virus to other pigs (Figure 4C,D). ELISA kits were used to detect antibodies to PRRSV N and GP5 in the experimental sera from both the experimental and control groups. The results revealed that PRRSV anti-N antibodies were present at 10 dpc and remained elevated throughout the trial, whereas they were not detected in the control group (Figure 4E). PRRSV anti-GP5 antibody was not detected in either the experimental or control group until the end of the trial (Figure 4F). The existence of this phenomenon may be related to PRRSV immune evasion.

### 3.8. Histopathology 

Through the HE staining results, we observed several changes in different organs between the experimental group and the control group. In the lungs, we observed widened alveolar septa, increased macrophages, lymphocyte exudation and hyperplasia of type II alveolar epithelial cells. Additionally, there were more neutrophils, macrophages, lymphocytes and detached necrotic cells in the alveolar lumina, indicating interstitial pneumonitis (Figure 5A). Moving on to the cortical region of the hilar lymph nodes, the experimental group showed a significant presence of erythrocytes and obvious lymphocyte necrosis. Furthermore, we observed more necrotic cell fragments and frequent karyopyknosis and karyorrhexis (Figure 5B). In terms of the myocardium, the experimental group displayed granular and vacuolar degeneration. Macrophages and lymphocytes were also observed oozing out around the endocardial vessels (Figure 5C). Moving to the spleen, the experimental group showed a significant reduction in the number of white marrow lymphocytes and the increased presence of erythrocytes (Figure 5D). Lastly, in the liver of the experimental group, we observed a large number of lymphocytes, macrophages and eosinophils exuding from the confluent area between the hepatic lobules (Figure 6A). In the experimental group, large hemorrhagic foci were observed in the medullary region of the kidneys. Additionally, there was an increased presence of macrophages and lymphocytes around the blood vessels (Figure 6B). Moving on to the brain, the experimental group showed signs of bruising. More lymphocytes and macrophages were observed within the blood vessels, and there was occasional exudation outside the blood vessels (Figure 6C). In contrast, the controls of the above-described tissues were all structurally clear, with regular cellular arrangement and no significant changes.

### 3.9. IHC Staining

The IHC results showed that positive signals of IHC were observed in the cytoplasm of macrophages, lymphocytes, type II epithelial cells and fine bronchial epithelial cells in the lungs of the experimental group (Figure 7A). Meanwhile, the H-score mean is 108. Moreover, we observed positive signals in the cytoplasm of lymphocytes and macrophages in the hilar lymph nodes of the experimental group (Figure 7B). Meanwhile, the H-score mean is 29. Furthermore, positive signals were observed in the cytoplasm of macrophages in the blood vessels and sinusoidal spaces of the liver in the experimental group as well as in the confluent area of perivascular exudate lymphocytes and macrophages (Figure 7C). Meanwhile, the H-score mean is 43. We also saw a positive signal in the cytoplasm of lymphocytes and macrophages in the spleen of the experimental group (Figure 7D). Meanwhile, the H-score mean is 230. Additionally, we found no significant positive signals in the myocardium, kidney and brain tissues of the experimental group (Figure 8A–C). In contrast, no positive signals were observed in the above-described tissues of pigs in the control group, and the mean H-score was 0. The positive signals described above all indicate the presence of the PRRS antigen.

## 4. Discussion

Since the first discovery of LNWK96 and LNWK130 in Liaoning Province, China, in 2017, NADC34-like strains have become endemic in several provinces or have become predominantly endemic in certain areas [11]. However, there have been no specific reports in Inner Mongolia. In this study, we successfully isolated and identified an NADC34-like strain from Tongliao City, Inner Mongolia. We named this strain CHNMGKL1-2304 and used it for follow-up studies. The strain was characterized through immunofluorescence, electron microscopy and PCR, which confirmed its successful isolation. Genome sequence analysis showed that the newly isolated strain had high homology with the JS2021NADC34 strain, reaching 97.7%. The homology of various ORF genes ranged from 94.7 to 99.3%. Additionally, the total genomic homology between this isolate and the LNWK96 strain (NADC34-like strain), which was the earliest strain found in China, was 92.2%. The homology of the ORF genes ranged from 85.2 to 96.2%, indicating that the NADC34-like strain has been constantly mutating since its introduction into China. We also found high homology between the different ORFs of the CHNMGKL1-2304 strain and the ORF nucleotides of the NADC34-like strains, particularly in the ORF2β and 3′UTR genes. The genome-wide homology ranged from 88.4 to 99.4%. Based on the analysis of the ORF5 gene and the genome-wide phylogenetic tree, we classified strain CHNMGKL1-2304 in sublineage 1.5 together with other NADC34-like strains. It was found to be closely related to strain JS2021 NADC34, suggesting a possible spread from Jiangsu to Inner Mongolia. Additionally, in the ORF5 gene phylogenetic tree, all isolated NADC34-like strains were grouped together on a large branch. Most of the NADC34-like strains from 2017 were also grouped together based on time in the genome-wide phylogenetic tree. Surprisingly, the strains from group 3 (isolated from 2018 to 2023) showed a gradual increase in size, indicating a unique epidemic trend and suggesting the popularity of NADC34-like strains. These strains have adapted well to localization. 

Since the NSP2 gene of PRRSV has the highest coefficient of variation and is considered the virulence gene, we compared the NSP2 gene of the CHNMGKL1-2304 strain with other NADC34-like strains in this study. We found a deletion of 100 aa (328–427 aa) in the NSP2 gene of NADC34-like strains relative to the VR2332 strain. The NADC34 strains have four deletion patterns at the NSP2 position—100 (328–427 aa), 100 + 8 (328–427 aa + 501–508 aa), 100 + 1 (328–427 aa + 483 aa/496 aa) and 100 + 4 (328–427 aa + 483–486 aa)—and all of these deletions occurred in the high variability area of NSP2 (Appendix A). It is believed that the discovery of these deletion sites can provide a basis for later studies of virulence genes. NSP2 is a key protein in viral pathogenesis, immunity and diagnostics, and the emergence of new PRRSV isolates often involves changes in NSP2 [20]. For instance, HP-PRRSV has a 30-amino-acid deletion in NSP2, while NADC30-like PRRSV has 131 discontinuous amino acid deletions in NSP2. These unique deletion patterns can be used to distinguish NADC34-like strains from other PRRSV lineages. The CHNMGKL1-2304 strain belongs to the NADC34-like family but differs slightly from other NADC34 strains in certain genomic regions and amino acid sites. The ongoing spread and recombination of PRRSV in nature pose an unpredictable threat to the pig industry. Recombination of PRRSV helps maintain genetic diversity among strains but also hinders the generation of neutralizing antibodies through vaccination, making it more challenging to defend against and control PRRSV. The LNWK96 and LNWK130 strains, first discovered in China, are recombinants derived from the IA/2014/NADC34 major parental strain and the ISU30 and NADC30 minor parental strains. These recombinants have shown abortion and mortality rates of 20% and 10%, respectively [21]. Meanwhile, the FJ0908 strain found in Fujian Province of China was also obtained by recombination of IA/2014/NADC34 as the major parental strain with ISU30 as the minor parental strain. This strain had a high abortion and mortality rate of 25% and 40%, respectively [22]. Therefore, recombination may lead to the development of more adaptive and pathogenic strains. In addition, CH/2018/NCV-Anhel-1, HLJZD30-1902 and HLJDZD32-1901 did not show any recombination events. To investigate the recombination of the CHNMGKL1-2304 strain, we performed recombination analyses using RDP (version 4) and Simplot (version 3.5.1) software. The results indicated that this isolate did not exhibit significant recombination with local strains. This suggests that the CHNMGKL1-2304 strain may not have undergone recombination with local strains or vaccine strains after spreading from Jiangsu to Inner Mongolia. Since its introduction to China from North America in 2012, the NADC30 strain has undergone extensive mutation and recombination, leading to its widespread epidemic in China between 2014 and 2016 [23]. Although we did not find any recombination of the CHNMGKL1-2304 strain with other PRRSVs in China, we cannot rule out the possibility of recombination in the future. In this study, we aimed to understand the pathogenicity of the NADC34-like strain isolated from Inner Mongolia. To achieve this, all piglets in the experimental and control groups were autopsied at the end of the pathogenicity experiment and evaluated for their pathogenicity. The results showed that the strain caused a gradual increase in the clinical scores of piglets in the experimental group, which tended to stabilize at 10 and 14 dpc. Moreover, the CHNMGKL1-2304 strain had a rapid onset of disease, with a rapid increase in body temperature and anorexia on the first day after infection. This was consistent with the onset of disease of its closely related strain, JS2021NADC34. In this study, we found that the challenged group consistently had a high fever from 7 to 14 dpc until the end of the trial, while the control group did not. Previous studies on the pathogenicity of the JS2021NADC34 strain showed that this strain caused a gradual increase in body temperature in the piglets of the experimental group after the challenge, followed by a gradual decrease after 7 dpc [14]. It shows that there is a difference in the duration of fever between the two strains. In contrast, the 2019 isolate of the HLJDZD32-1901 strain was found not to cause fever symptoms in piglets in a pathogenicity experiment [24]. It can be seen that different NADC34 strains cause varying degrees of fever in piglets. We also observed that the weight of piglets in the experimental group increased slowly after being challenged, averaging 0.196 kg/day, while the control group grew by 0.4 kg/day. Meanwhile, the daily weight gain of HLJDZD32-1901 PRRSV-infected pigs exceeded 0.2 kg/day [24]. In contrast, JS2021NADC34 PRRSV-infected pigs gained more than −0.4 kg/day [14]. Additionally, the average weight gain of pigs infected with the PRRSV-ZDXYL-China-2018-1 strain was −0.22 kg/day [15]. This demonstrates that the NADC34-like strain significantly affects the growth cycle of the piglets and the timing of farrowing, leading to significant economic losses. Moreover, one of the three piglets in the experimental group developed walking disorders and became paralyzed three days after the challenge. No piglets died during the entire trial period. Meanwhile, there were no deaths in pigs infected with the HLJDZD32-1901 and PRRSV-ZDXYL-China-2018-1 strain [15,24]. However, the JS2021NADC34 strain found in Jiangsu Province showed that two of the four pigs died on day 8 and one on day 10 even though they used 2-month-old piglets, whereas this study used 1-month-old piglets [14]. The current study shows that the NADC34-like strains found in China have different mortality and miscarriage rates [11]. Epidemiological investigations of the affected farms revealed that the strain caused piglet mortality of approximately 10%, whereas in the present study, only three piglets were placed in the challenge group, resulting in no piglet deaths. The most common symptom of PRRSV infection in pigs is persistent viraemia [25]. In contrast, in this study we found that the challenge group developed typical viraemia at 3 dpc, which persisted until the end of the trial. Some studies have shown that viraemia can develop within 12 h after PRRSV infection in pigs [26]. However, samples were not collected for experimentation during that time period in this study.

Studies have shown that the PRRSV-1 Lena strain and the PRRSV-2 JXwn06 strain have viral loads of approximately 10^7^ copies/mL in the lungs and lymph nodes [27,28,29]. Meanwhile, the viral load in lungs and lymph nodes of piglets infected with the HLJDZD32-1901 strain exceeded 2.5 × 10^7^ and 10^6^ copies/g [24], whereas in this study, the tissues with higher viral loads as quantified were the lungs and hilar lymph nodes, which reached 10^7.18^ and 10^8.19^ copies/µL, respectively. This result is consistent with the visual observation of the severity of the lesions. Furthermore, nucleic acid testing of nasal and stool swabs revealed that the challenge group was able to contaminate the environment at 3 dpc, resulting in infection of other piglets. This finding provides evidence that the NADC34-like strain is excreted into the environment via nasal and stool swabs. Additionally, we found that all anti-N antibodies to PRRSV turned positive at 10 dpc and gradually increased, whereas anti-GP5 antibodies were not produced until the end of the experiment. Meanwhile, piglets infected with the HLJDZD32-1901 and JS2021NADC34 strains tested positive for antibodies to anti-N at 7 dpc [14,24]. Studies have shown that N is the most immunogenic viral protein, but antibodies to the anti-N protein are non-neutralizing and appear earlier in the presence of PRRSV infection [9,14,30]. However, the GP5 protein of PRRSV is a structural protein that induces neutralizing antibodies. There is a delay in the appearance of antibodies to the anti-GP5 protein (after 21 or 28 days of immunization), which is a reflection of PRRSV immunosuppression [31,32,33].

We observed typical pathological changes caused by PRRSV in the tissues of the experimental group and found significant positive signals in the cells in IHC of the lungs, lymph nodes, liver and spleen. Through HE staining and IHC, we discovered that the pathological changes caused by the CHNMGKL1-2304 strain were more severe than those of the ZDXYL-China-2018-1 and HLJDZD32-1901 strains. In addition, we observed that the myocardium of the experimental group exhibited granular and blistering degeneration, resulting in a visibly floppy texture. This phenomenon was attributed to pulmonary solid changes in the lungs of the piglets.

In summary, the CHNMGKL1-2304 strain isolated in this study is a moderately virulent strain. This strain was more pathogenic than the HLJDZD32-1901 and PRRSV-ZDXYL-China-2018-1 strains but weaker than the JS2021NADC34 strain. This study systematically evaluates the pathogenicity of the newly discovered NADC34-like strain in Inner Mongolia and provides a basis for epidemiological and subsequent studies of PRRSV.

## Figures and Tables

**Figure 1 viruses-16-00683-f001:**
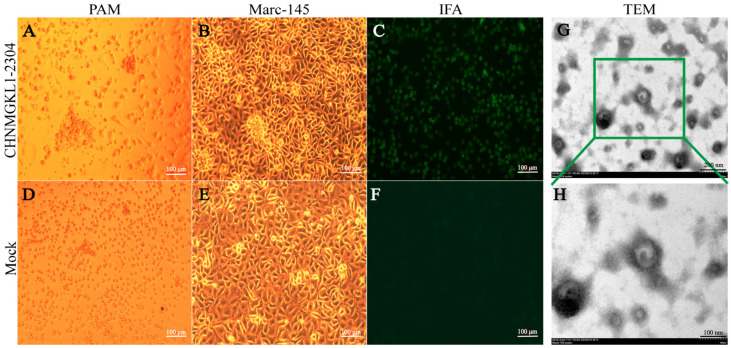
The CHNMGKL1-2304 strain was able to induce lesions on PAM and Marc-145 cells. The presence of viral particles was determined through IFA and electron microscopy. Above is the infection group. Mock group is shown below. (**A**) Infection of PAM cells by the CHNMGKL1-2304 strain causes lesions. (**B**) Infection of Marc-145 cells by the CHNMGKL1-2304 strain causes lesions. (**C**) IFA results of PAM cells infected with CHNMGKL1-2304 strain. (**D**–**F**) Control Group. (**G**) Virus particles from the CHNMGKL1-2304 strain propagated in PAM cells. Phosphotungstic-acid-stained PRRSV particles of 50–70 nm in diameter is visible. Scale bar = 200 nm. (**H**) Enlarging the field of view within the frame. Scale bar = 100 nm.

**Figure 2 viruses-16-00683-f002:**
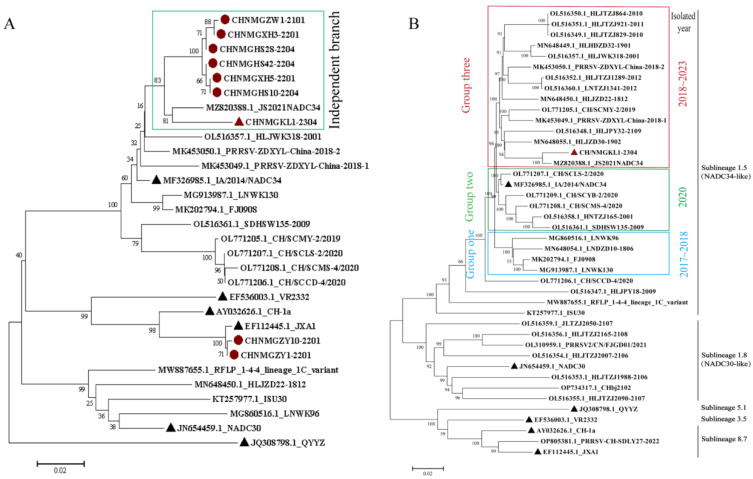
Phylogenetic trees constructed based on ORF5 gene and whole genome. (**A**) Phylogenetic tree based on ORF5 gene. (**B**) Phylogenetic tree based on whole genome. The phylogenetic tree was constructed using the distance-based neighbor-joining algorithm in Mega7, with 1000 bootstrap replicates (
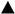
 indicates the reference virus. 
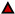
 and 
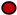
 indicate the strain isolated from Inner Mongolia. Blue box: group one. Green box: group two. Red box: group three).

**Figure 3 viruses-16-00683-f003:**
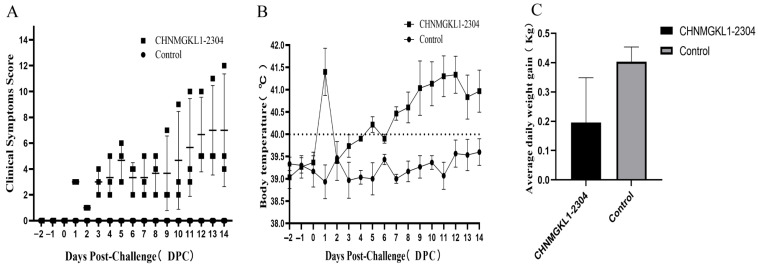
Results of clinical signs in piglets infected with the CHNMGKL1-2304 strain. (**A**) Results of the clinical assignment of values. (**B**) Results of body temperature changes. (**C**) Results of average daily weight gain. Graphs were made using GraphPad Prism 8 software.

**Figure 4 viruses-16-00683-f004:**
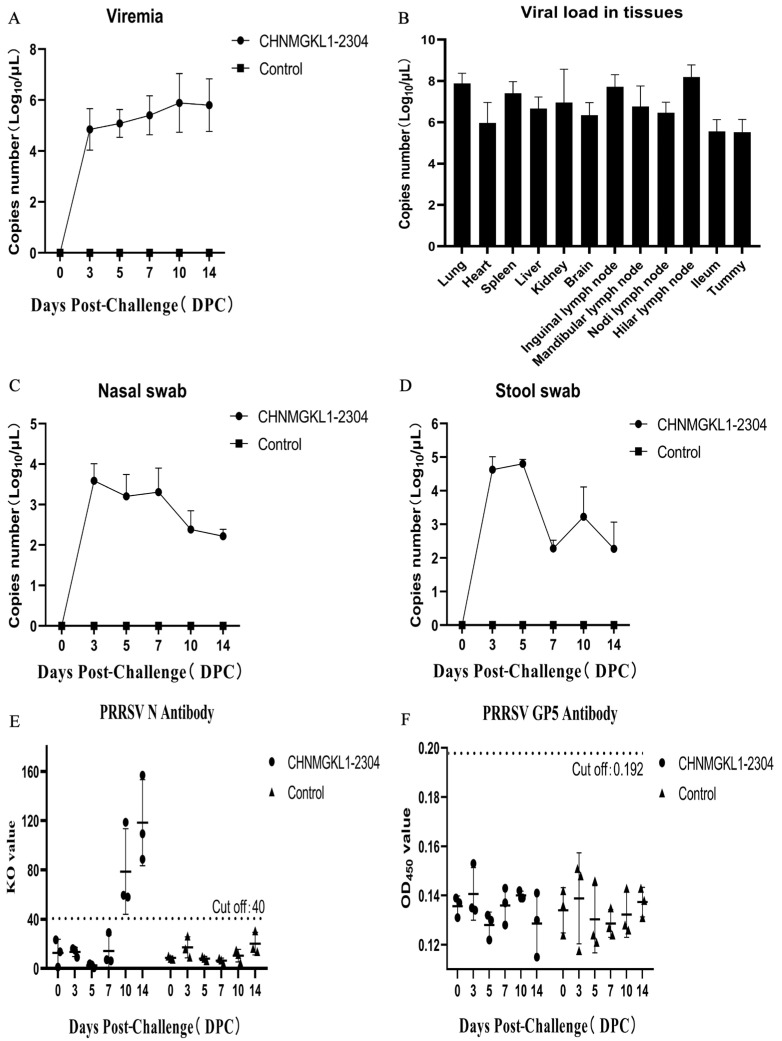
Viral load and antibody profile in the piglets infected with CHNMGKL1-2304 virus. (**A**,**B**) The copy number of virus genome was determined in the blood and tissues of piglets after challenge using real-time fluorescence quantitative PCR. (**C**,**D**) The copy number of virus genome was identified in nasal and stool swabs of piglets after challenge using real-time fluorescence quantitative PCR. (**E**,**F**) The antibody titer of N and GP5 in the sera from the experimentally challenged piglets was determined using commercial ELISA kits. Graphs were generated using GraphPad Prism 8 software.

**Figure 5 viruses-16-00683-f005:**
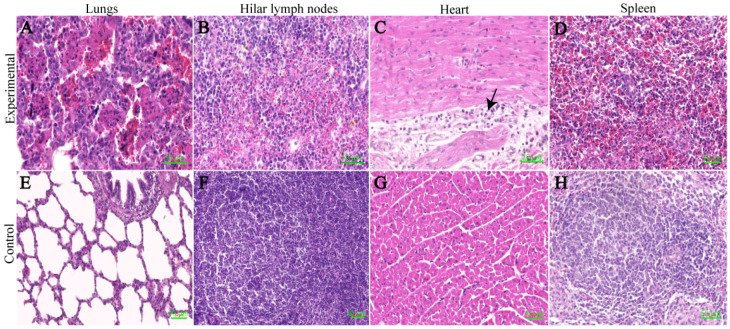
Results of tissue hematoxylin–eosin staining. (**A**) Lungs: A typical indirect pneumonia with type II alveolar epithelial cell hyperplasia is seen, along with a high number of neutrophils, macrophages, lymphocytes and detached necrotic cells. (**B**) Hilar lymph nodes: A large number of erythrocytes are seen in the cortical area, and lymphocyte necrosis is evident. (**C**) Heart: Myocardial granules are vesicularly degenerated, and macrophages and lymphocytes are seen oozing around the endocardial vessels at the black arrows. (**D**) Spleen: There is a marked decrease in the number of white marrow lymphocytes, and a large number of red blood cells are seen. (**E**–**H**) Control group. Use AI software to make drawings. Scale bar = 50 μm.

**Figure 6 viruses-16-00683-f006:**
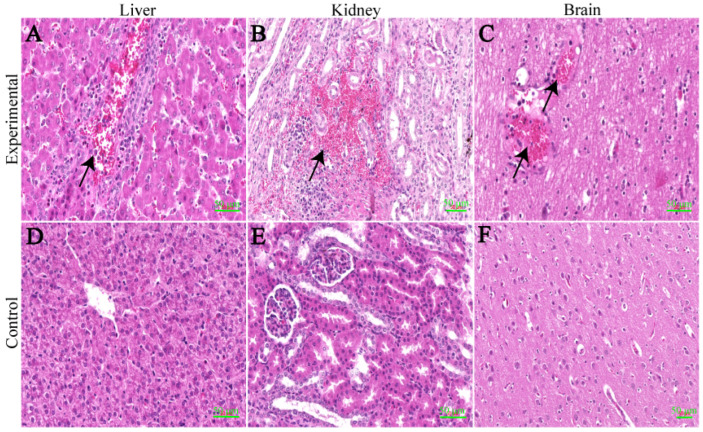
Results of tissue hematoxylin–eosin staining. (**A**) Liver: More lymphocytes, macrophages and eosinophils are seen oozing from the confluent area. (**B**) Kidney: Larger foci of hemorrhage are seen in the medullary region, and more macrophages and lymphocytes are seen oozing from the perivascular area. (**C**) Brain: Higher numbers of lymphocytes and macrophages are seen within the vessels, and occasional extravasation is seen. (**D**–**F**) Control group. Black arrows indicate lesion areas. Scale bar = 50 μm.

**Figure 7 viruses-16-00683-f007:**
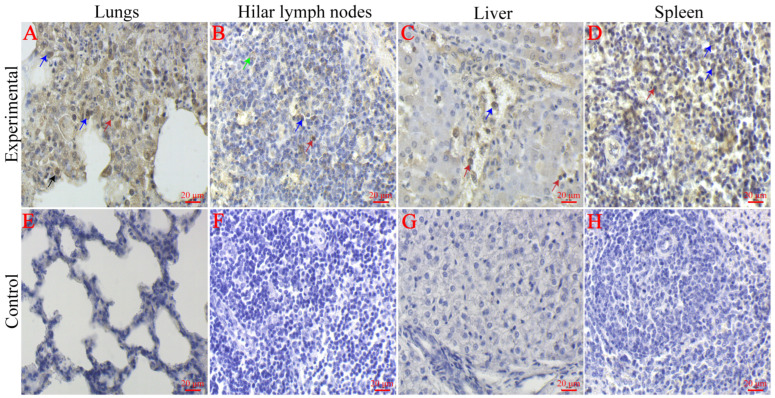
IHC results of tissues of piglets infected with CHNMGKL1-2304 strain. (**A**,**E**) Lungs. (**B**,**F**) Hilar lymph nodes. (**C**,**G**) Liver. (**D**,**H**) Spleen. We use different-colored arrows to indicate the types of cells that are infected with PRRSV. Red arrows: lymphocytes. Black arrows: type II alveolar epithelial cells. Blue arrows: macrophages. Green arrows: reticulocytes. Scale bar = 20 μm.

**Figure 8 viruses-16-00683-f008:**
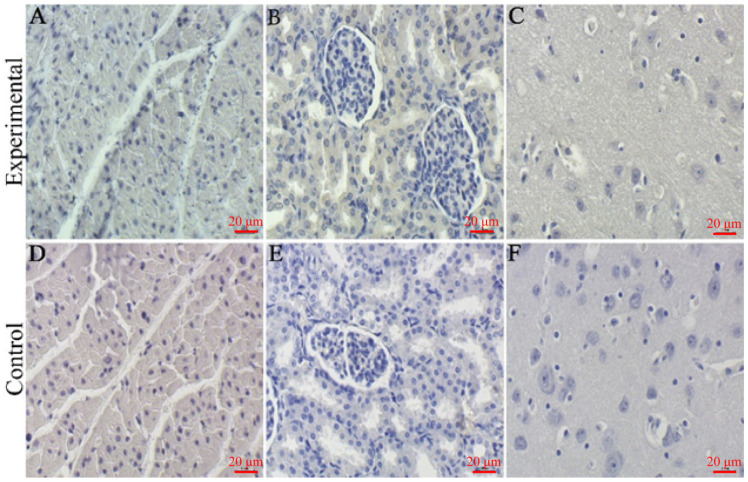
IHC results of CHNMGKL1-2304 strain after infection of piglets. (**A**,**D**) Heart. (**B**,**E**) Kidney. (**C**,**F**) Brain. Scale bar = 20 μm.

**Table 1 viruses-16-00683-t001:** Primers for amplicating the genome of CHNMGKL1-2304 strain.

Primer Name	Sequences (5′–3′)	Annealing Temperature (°C)	Size (bp)
5′UTR-A	F:ATGACGTATAGGTGTTGGCTC	52	335
	R:CAAGCCCAACACTCCAAG		
34-1-B	F:GCACCTTGCTTCTGGAGT	54	2591
	R: GCGTTGTAGTTGTTAGTTTCG		
34-2-C	F:TGCCAGATTGTAAGCCCGTCCCT	57	2646
	R:ACCAGCGTAACCAGCAAGGAA		
34-3-D	F: CCCGTTTGCCGTTCCTGGTT	56	2350
	R: TGCGTCCGTGTTGTCGTG		
34-4-E	F: CCCGTCGGCAGTATCTTT	54	2758
	R: AAGACTTCGTCCAGAGGG		
34-5-F	F: GAACTTGTGGTTGGGATG	54	2758
	R: TGCTCAGAGTGAACGGTAG		
34-6-G	F: GGGAAGATTACAATGACGC	53	2573
	R: TTAAGAGGTGCAAGAGCC		
34-7-H	F: GCGGAACAATGGGGTCGT	55	1047
	R: AACCATGCGGCCGTAATTAA		

**Table 2 viruses-16-00683-t002:** Homology between different lineages (%).

	Sublineage 1.5		Sublineage 1.8	Sublineage 3.5	Sublineage 5.1	Sublineage 8.1
Nucleotides	JS2021NADC34	IA/2014/NADC34	NADC30	QYYZ	VR2332	Ch-1R
Complete genome	97.7	94.9	84.9	81.2	83.1	83.1
5′UTR	94.7	93.6	91.5	92.0	89.5	89.4
Nsp1	98.7	94.5	84.8	81.3	83.8	82.9
Nsp2	97.0	92.1	75.1	71.1	73.7	74.7
Nsp3	98.5	96.1	84.9	79.6	83.7	81.8
Nsp4	98.4	96.4	81.4	79.7	82.7	83.2
Nsp5	96.9	92.2	83.5	79.8	84.3	83.7
Nsp6	97.9	95.8	89.6	85.4	91.7	85.4
Nsp7	97.2	93.3	81.5	78.9	81.6	80.1
Nsp8	97.0	96.3	90.4	88.9	89.6	88.9
Nsp9	98.2	96.0	88.6	85.6	86.8	86.6
Nsp10	98.6	96.0	90.2	84.5	85.7	85.8
Nsp11	99.0	96.4	85.3	85.6	85.3	87.1
Nsp12	97.8	95.5	83.3	83.5	82.6	82.1
ORF2a	96.6	95.1	85.7	86.1	86.9	87.0
ORF2b	96.8	95.9	87.8	91.0	89.2	88.7
ORF3	96.5	94.8	85.0	82.1	83.4	83.0
ORF4	97.8	94.6	93.1	84.4	86.4	85.5
ORF5	97.0	95.9	86.9	83.1	85.7	85.7
ORF6	97.9	97.7	93.5	88.8	88.6	87.6
ORF7	96.2	96.0	94.1	87.6	89.8	90.9
3′UTR	99.3	99.3	96.7	90.1	88.0	88.2

**Table 3 viruses-16-00683-t003:** Homology between CHNMGKL1-2304 and other strains within same sublineage (%).

Nucleotides	LNWK96	LNWK130	FJ0908	CH/SCMY-2/2019	HLHDZD32-1901	HLJPY18-2009	HLJZD30-1902	CH/SCLS-2/2020	PRRSV-ZDXYL-CHINA-2018-1	PRRSV-ZDXYL-China-2018-2
Complete genome	92.2	92.8	93.3	93.2	93.5	88.4	94.3	94.4	94	93.8
5′UTR	91.0	94.0	91.5	93.6	93.6	89.9	93.0	93.1	93.6	93.1
Nsp1	89.8	89.8	91.8	93.0	93.0	80.7	94.1	94.3	93.0	92.9
Nsp2	88.9	88.6	89.4	91.3	90.3	72.3	91.0	91.0	91.3	90.9
Nsp3	93.6	93.3	94.2	94.8	94.5	91.0	95.1	96.1	94.8	94.4
Nsp4	95.3	94.1	95.1	96.2	95.8	95.1	95.9	96.1	96.2	95.6
Nsp5	90.8	91.6	91.0	91.6	90.6	90.6	91.6	91.6	91.6	90.2
Nsp6	95.8	95.8	93.8	97.9	93.8	97.9	93.8	95.8	97.9	95.8
Nsp7	92.4	93.2	93.4	93.3	91.0	92.4	93.3	92.9	93.3	92.1
Nsp8	96.3	93.3	94.8	94.1	95.6	96.3	95.6	95.6	94.1	94.8
Nsp9	94.8	94.7	95.1	94.9	94.9	89.3	95.6	96.0	94.9	94.8
Nsp10	95.0	95.2	94.9	95.2	94.7	93.3	95.7	96.0	95.6	95.3
Nsp11	95.5	96.0	95.8	94.0	95.7	95.7	96.3	96.1	95.4	96.3
Nsp12	94.4	94.2	94.2	89.6	94.6	94.0	94.8	95.0	95.0	94.0
ORF2α	94.2	94.3	94.6	94.0	93.6	93.4	94.2	94.7	93.6	93.9
ORF2β	94.6	94.6	95.5	95.5	95.0	95.5	95.0	95.5	95.0	95.5
ORF3	89.9	92.2	92.2	92.3	93.5	93.1	94.6	94.4	94.1	94.0
ORF4	91.2	92.4	92.2	89.8	93.9	93.5	95.0	94.4	94.4	95.0
ORF5	85.2	94.2	94.4	91.9	95.0	95.2	95.5	91.7	94.4	95.9
ORF6	93.3	95.8	95.8	91.0	96.6	97.1	96.2	97.1	95.6	95.8
ORF7	91.7	93.3	93.5	92.7	94.9	93.3	95.4	95.2	95.2	94.9
3′UTR	96.0	97.1	98.7	94.7	96.1	98.0	98.0	100.0	98.5	96.7

## Data Availability

All the data that support the findings of this study are available from the corresponding author upon reasonable request.

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
