# Peer review of "Evolution Characterization and Pathogenicity of an NADC34-like PRRSV Isolated from Inner Mongolia, China"

_viruses, 2024, doi:10.3390/v16050683_

Round 1
Reviewer 1 Report
Comments and Suggestions for Authors
Zhao et al described the evolution characterization and pathogenicity of NADC34-like strain isolated from Inner Mongolia. This is a great research article about PRRSV NADC34-like strain. Authors give overall goodbackground of NADC34-like PRRSV in China. This manuscript is suitable for publish in Viruses after minor revision.
Major commons:
1. There are no table in the text, please insert tables into manuscript. 2. Title: Evolution characterization and pathogenicity of NADC34-like porcine reproductive and respiratory syndrome virus isolated from Inner Mongolia. Comments on the Quality of English LanguageZhao et al described the evolution characterization and pathogenicity of NADC34-like strain isolated from Inner Mongolia. This is a great research article about PRRSV NADC34-like strain. Authors give overall goodbackground of NADC34-like PRRSV in China. This manuscript is suitable for publish in Viruses after minor revision.
Major commons:
1. There are no table in the text, please insert tables into manuscript. 2. Title: Evolution characterization and pathogenicity of NADC34-like porcine reproductive and respiratory syndrome virus isolated from Inner Mongolia.Author Response
Please see the attachment.

Reviewer 2 Report
Comments and Suggestions for Authors
The manuscript reports a pathogenicity investigation on a recent PRRSV strain. The experimental design is correct. Nevertheless, the methods should be detailed and clarified and the novelty, the scientific significance of results should be emphasized and highlighted to improve the quality of the manuscript.
Major points:
There have not been any Tables added to the m.s. I missed Tables 1; 2 and 3.
lines 39-40: blue ear disease: add reference!
Add a single, but informative sentence about the virus itself and its genome! E.g.: Arteriviridae; +ssRNA; genome structure etc.
Chapter 2: The Methods should be more clarified!!! Now the methods are not repeatable.
Chapter 2.4.: More details of the RT-PCR are needed: Table1 with primers; reaction mix, thermal cycles, etc.
line 100: Do you mean intramuscular and intranasal challenge? How was the intranasal administration carried out?
line 101: What was the volume?
line 107: Which kit?
lines 109-110: More details of RT-qPCR!!!!
chapter 2.8.: Was it an ELISA test or not? You did NOT measure "antibody changes". In the M&M chapter write down how you measured the antibody quantities/titres, and if there will be changes, report them in Results and discuss it in Discussion.
The calculation of sequence homology, recombination analysis etc. are missing from the M&M chapter, while their results are mentioned in chapter 3.3.
Caption of Fig1: Write some words what can we see! Implement some info from lines 128-134!
Enlarge the figures and the labels on them to make them easy to read!
In results and discussion explain why cannot we see changes of anti-GP5 antibody, while anti-N could be seen!
There is a lot of information and picture on histopathology and IHC, but there is not quantified data! Try to use a scoring scale or any similar techniques to compare the two groups!
In Discussion there is a lot of data on genetic comparisons, but its meaning, pathological aspects are not discussed properly. Reduce the meaningless comparisons of pure % values, but discuss the pathology!
The comparisons would be more meaningful, if there would be another pathogenic strain involved in the same study in a comparative way. In the lack of this type of data, please refer to pathogenicity of other strains from other studies to make the results comparable!
Minor points:
The title lacks the virus and the country. Please change, like this: "Evolution Characterization and Pathogenicity of a of NADC34-like PRRSV Strain Isolated from Inner Mongolia (China)"
lines 15-16: "porcine/swine" instead of livestock
line 27: "nasal", not "Nasal"; "brownish positive signals" - NOT scientific, specify!
lines 50-51: "never been reported", but there is two references! What do you mean with that?
Use the phrases"anti-N" and "anti-GP5" antibodies!
Comments on the Quality of English LanguageRe-editing and rephrasing is needed to simplify the long sentences.
Check twice the typos!
Round 2
Reviewer 2 Report
Comments and Suggestions for Authors
The authors corrected and improved the quality of the m.s. according to the reviewer's advice.